# Evidence of a Recent Bottleneck in *Plasmodium falciparum* Populations on the Honduran–Nicaraguan Border

**DOI:** 10.3390/pathogens10111432

**Published:** 2021-11-04

**Authors:** Alejandra Pinto, Osman Archaga, Ángel Mejía, Lenin Escober, Jessica Henríquez, Alberto Montoya, Hugo O. Valdivia, Gustavo Fontecha

**Affiliations:** 1Microbiology Research Institute, National Autonomous University of Honduras, Tegucigalpa 11101, Honduras; mpinto@unah.edu.hn (A.P.); osman.archagav@gmail.com (O.A.); fernando.mejia@unah.edu.hn (Á.M.); 2National Malaria Laboratory, National Department of Surveillance, Ministry of Health of Honduras, Tegucigalpa 11101, Honduras; lenonarturo@yahoo.com (L.E.); henriquezjessica20@gmail.com (J.H.); 3National Center for Diagnosis and Reference, Health Ministry, Managua 11001, Nicaragua; parasitologia@minsa.gob.ni; 4Department of Parasitology, U.S. Naval Medical Research Unit No, 6 (NAMRU-6), Lima 07006, Peru; hugo.o.valdivia.ln@mail.mil

**Keywords:** malaria, *Plasmodium falciparum*, Honduras, Nicaragua, *pfmsp*-1, *pfmsp*-2, genetic diversity

## Abstract

The countries of Central America and the island of Hispaniola have set the goal of eliminating malaria in less than a decade. Although efforts to reduce the malaria burden in the region have been successful, there has been an alarming increase in cases in the Nicaraguan Moskitia since 2014. The continuous decrease in cases between 2000 and 2014, followed by a rapid expansion from 2015 to the present, has generated a potential bottleneck effect in the populations of *Plasmodium* spp. Consequently, this study aimed to evaluate the genetic diversity of *P. falciparum* and the decrease in allelic richness in this population. The polymorphic regions of the *pfmsp*-1 and *pfmsp*-2 genes of patients with falciparum malaria from Honduras and Nicaragua were analyzed using nested PCR and sequencing. Most of the samples were classified into the K1 allelic subfamily of the *pfmsp*-1 gene and into the 3D7 subfamily of the *pfmsp*-2 gene. Despite the low genetic diversity found, more than half of the samples presented a polyclonal K1/RO33 haplotype. No sequence polymorphisms were found within each allelic subfamily. This study describes a notable decrease in the genetic diversity of *P. falciparum* in the Moskitia region after a bottleneck phenomenon. These results will be useful for future epidemiological investigations and the monitoring of malaria transmission in Central America.

## 1. Introduction

The incidence of malaria in the Americas has decreased by 53% in the last decade, reaching 0.9 million cases in 2019 [1]. However, progress in malaria elimination is not homogeneous for all countries. In this regard, 86% of malaria cases on the continent are reported by three countries: Brazil, Colombia and Venezuela, and the latter country reported almost half a million cases in 2019, representing a 13-fold increase since 2000 [1]. Although the Central American isthmus is responsible for only 2.4% of cases in the Americas, there are also notable differences in the epidemiology of malaria among the seven Central American countries. For example, Costa Rica, Panama and Nicaragua reported increases in case incidence of more than 40% in 2020 compared to 2015, while Belize, Guatemala and Honduras have met the goals set for 2020 by the Global Technical Strategy 2016–2030 to reduce incidence by at least 40% [2]. On the other hand, Belize and El Salvador have reported zero cases of malaria during the last years [1]. 

Currently the Central American region most affected by malaria is the Moskitia, which comprises the easternmost region of Honduras and the northwestern region of Nicaragua. The Moskitia is a region shared by both countries with little human development, scarce communication routes and predominantly indigenous populations with their own cultural characteristics and little access to health services. The incidence of malaria in Honduras and Nicaragua reached its lowest point in 2014 with around 1163 cases, and the downward trend has remained relatively constant in Honduras [1]. However, the number of malaria cases in Nicaragua exceeded 25,000 in 2020, returning to the figures reported in 2000 (Figure 1) [1]. Another unprecedented phenomenon in the region is the change in the proportion of *Plasmodium falciparum* malaria cases, which in Moskitia had never exceeded 20%, while now exceeds 50% on the Nicaraguan side. These changes in malaria transmission patterns and a recent bottleneck phenomenon are very likely to influence parasite populations and their genetic diversity.

The genetic diversity of *Plasmodium* spp. has been related to the ability of parasites to adapt adequately to their hosts through the selection of advantageous traits, such as antigenic variability that allows them to evade the immune response, as well as the emergence of mutations responsible for resistance to antimalarials [3,4,5,6,7]. Genetically diverse populations of *Plasmodium* are common in regions with high transmission patterns, such as sub-Saharan Africa and some Southeast Asian countries [8,9,10,11,12], but as the incidence of malaria declines, the parasite’s genetic diversity is expected to decline as well [13,14]. Central America is considered a region of moderate to low malaria transmission, and studies on the genetic diversity of *Plasmodium* spp. in Central America are scarce. Most studies have analyzed polymorphic coding genes (*P. falciparum* glutamate rich protein *pfglurp*, merozoite surface proteins *pfmsp*-1 and *pfmsp*-2, *P. vivax* apical membrane antigen *pvama*-1, merozoite surface proteins *pvmsp*-1, *pvmsp*-142, circumsporozoite protein *pvcsp*, and the surface protein *pvs48*/*45*) to assess the diversity of *P. falciparum* and/or *P. vivax* in some countries of the isthmus [14,15,16,17,18,19], whereas other studies have used microsatellite markers [20,21] or next generation sequencing using a selectively amplified whole genome approach (sWGA) [22]. Despite differences in the methods, most of these studies conclude that the parasite populations in Central America present a low diversity in relation to parasites of Sub-Saharan Africa [23]. This is much less clear in the Americas, where some of the regions with high incidence of malaria are still within the low to medium transmission levels and accordingly show low diversity of the parasite populations [24].

Some of the most widely used markers are two genes that encode the merozoite surface proteins of *P. falciparum* (*pfmsp*-1 and *pfmsp*-2) [17,25,26,27,28,29]. These markers are very informative due to their high rate of size and sequence polymorphisms, due in part to their tendency to recombine during the sexual phase of the cycle [30,31]. Furthermore, both markers make it possible to detect polyclonal infections. The *pfmsp*-1 gene is located on chromosome 9 and is made up of 17 sequence blocks, of which block 2 is particularly polymorphic [32]. This gene segment allows the parasite to be classified into three allelic subfamilies: K1, MAD20 and RO33, and to detect alleles according to the size of the amplified fragments. The *pfmsp*-2 gene is located on chromosome 2. It is composed of five sequence blocks, of which the most polymorphic is the third block, allowing the parasites to be classified into two allelic subfamilies: FC27 and 3D7 [27,29,33,34]. This study aimed to analyze the changes in the genetic diversity of *P. falciparum* population circulating in the Moskitia after a recent bottleneck phenomenon.

## 2. Results

A total of 160 blood samples were collected between 2018 and 2021 with a microscopic diagnosis of *P. falciparum* malaria confirmed by PCR. Block 2 of the *pfmsp*-1 gene was amplified, and 147 (92%) samples were positive for at least one of the three allelic subfamilies. The cause of the lack of amplification of the remaining 13 samples is unknown. The overall frequency of K1, MAD20 and RO33 subfamilies was 84.4%, 23.8% and 61.2% respectively. Block 3 of the *pfmsp*-2 gene was also analyzed, and 55 (34%) samples amplified for at least one of the subfamilies 3D7 or FC27. The 3D7 allelic subfamily was present in 92.7% of the samples, while the FC27 subfamily was detected in the remaining 7.3% (Table 1) (Appendix A).

The alleles of each subfamily were classified according to their approximate size in base pairs on a 2% agarose gel. A single allele was found for the subfamilies K1 (235 bp), RO33 (335 bp), 3D7 (276 bp) and FC27 (327 bp). Three alleles were found for MAD20 (146 bp, 183 bp, and 202 bp) (Figure 2).

The minority of samples showed monoclonal infections with a single allele. Thirty-three (22.5%) samples amplified only K1, 16 (10.9%) samples amplified only MAD20 and three (2.04%) samples amplified only RO33. The number of samples with polyclonal infections, that is, that showed two or more alleles, was 96 (65.3%) for the *pfmsp*-1 gene. In contrast, no polyclonal infections were detected for *pfmsp*-2. The different combinations of alleles of *pfmsp*-1 are shown in Table 2. No intrafamily mixed infections were detected. Regarding the interfamily mixed infections, the most frequent combination of alleles was K1 and RO33 (52.4%). Five samples (3.4%) revealed the simultaneous presence of alleles for the three subfamilies of *pfmsp*-1 (Figure 3).

As shown in Figure 4 and Table 2, the relative distribution of alleles in the samples collected in Honduras and Nicaragua are not statistically significant. According to Pearson’s Chi square test, the *p*-values were 0.2296 for K1, 0.9542 for MAD20, and 0.4960 for RO33. Consequently, the parasites in the Moskitia region appear to behave as a homogeneous population.

To confirm the correct identity of the amplicons and detect sequence polymorphisms within each of the subfamilies, 21 amplicons from K1, 16 from MAD20, 10 from RO33, four from 3D7 and one from FC27 were successfully sequenced and assembled. According to the results retrieved by the NCBI BLAST, all the amplified sequences corresponded to the *pfmsp*-1 and *pfmsp*-2 genes. The sequences obtained in this study were deposited in Genbank with the accession numbers MZ723284–MZ723293 for K1, MZ723294–MZ723308 for MAD20, MZ723309–MZ723313 for RO33, MZ747207 for 3D7, and MZ747208 for FC27.

No polymorphisms were detected between the sequences within each marker, as can be seen in the chromatogram alignment of Figure 5. Consequently, a single allele of K1, RO33, 3D7 and FC27 was found. The only polymorphisms detected were among the three alleles of the MAD20 subfamily. The nucleotide sequences of the five markers (K1, MAD20, RO33, 3D7 and FC27) were translated in silico, and the amino acid sequences were determined. Figure 6 shows the tripeptide repeat sequences of K1, MAD20, RO33, and 3D7, as well as the amino acid sequence of FC27. These sequences were analyzed by the Protein Blast tool and the results indicated identity percentages of 93.2% for RO33 and 100% for K1, MAD20 and 3D7, with respect to sequences previously deposited in the database of isolates of various geographic regions.

## 3. Discussion

The genetic diversity of *P. falciparum* populations is directly proportional to the intensity of malaria transmission in a geographic region [8]. High genetic diversity of the parasite is characteristic of areas of highest malaria transmission while diversity is expected to be minimal in low transmission settings. For example, a recent study conducted in Northeast India using merozoite surface protein 1 and 2 (*pfmsp*-1, *pfmsp*-2) and glutamate rich protein (*pfglurp*) revealed extensive parasite diversity in a state that contributes almost half of malaria cases to the region [28]. An earlier study demonstrated dramatic variation in geographic diversity and differentiation in different regions using microsatellite analysis. The mean heterozygosities in South American countries were less than half of those observed in Africa, with intermediate heterozygosities in the Asia/Pacific region [35].

Central America is considered a low transmission scenario, and although there are active foci in Panama, Costa Rica and Guatemala, the Moskitia region, shared between Honduras and Nicaragua, is responsible for the largest number of malaria cases in the Central American isthmus [1]. The number of malaria cases in the Moskitia had been declining continuously from 2000 to 2009, the year in which Nicaragua reported only 640 cases, representing a reduction of more than 97% since 2000. Even though the reduction of cases in Honduras has remained relatively constant to date, malaria in the Nicaraguan Moskitia has exceeded 25,427 cases in 2020, delaying two decades of progress in the fight against malaria.

Another important change in the epidemiology of malaria in the Moskitia is the increase in the proportion of cases caused by *P. falciparum*. Historically, the largest contributor to malaria cases was *P. vivax*, and falciparum malaria did not exceed 20%; however, now it contributes more than 28% of malaria cases in Honduras and more than 50% in Nicaragua. This change in the transmission pattern added to the possible bottleneck effect experienced by the populations of the parasite in the last decade makes it necessary to study the effects on the genetic diversity of the parasite.

In this study, the allele frequencies of the *pfmsp*-1 and *pfmsp*-2 genes were assessed. Both genes encode antigens with highly polymorphic regions that have been used extensively to study the epidemiology of malaria [27,28,36]. The most frequent allelic subfamily of *pfmsp*-1 was K1 (84.4%) followed by RO33 (61.2%) and MAD20 (23.8%). An earlier study using 30 samples collected from six municipalities in Honduras (two in the Moskitia region) between 2010 and 2011 found that K1 was the most prevalent subfamily (57%), followed by MAD20 (25%) and RO33 (17%) [17]. The authors reported that most alleles were evenly represented, and no differences were found between the Miskito region and the rest of the country. The study by López et al. included a broader geographic region of the country compared to other studies from the same period and despite having analyzed only 30 isolates of *P. falciparum*, it is the only one that reports high genetic diversity in the parasite population in Honduras [17]. Another previous study analyzed 56 samples collected between 1995 and 1996 in one municipality in Honduras (Tocoa, Colón) and found that the most frequent allelic subfamily was MAD20 (73.2%) followed by K1 (46.4%). RO33 was not described on that occasion [15]. These discrepancies can be attributed to the population dynamics of the parasite over time, or to the difference in the geographic areas where the samples were collected.

Based on extensive global data collection by Lê et al. [29], subfamilies K1 and MAD20 are the dominant types worldwide. In two Latin American countries (Peru and Brazil), K1 shows a slight predominance over MAD20, while the frequency of RO33 is only marginal. Similarly, a study carried out with samples from Colombia reports an almost absolute predominance of MAD20 and did not report the presence of RO33 [37]. It should be noted then that the frequency of RO33 in Central America has increased compared to previous reports, resembling the results of similar studies carried out in Africa and Asia [29,38].

Regarding the *pfmsp*-2 gene, our results reveal a clear predominance of the 3D7 subfamily (92.7%) over FC27 (7.3%). This result coincides with what was reported 10 years earlier in Honduras by López et al. [17] with 91.7% predominance of 3D7 in 12 samples analyzed. Studies of the genetic diversity of the *pfmsp*-2 gene in the Americas are scarce. Two studies conducted in Colombia and Brazil reported that both FC27 and 3D7 allelic subfamilies were found in almost the same proportions [37,39]. On the other hand, the study by Haddad et al. [15] carried out with samples from one municipality in Honduras, reported a clear predominance of FC27 over 3D7.

The most surprising result of the present study is the low genetic diversity found in the parasite population. All 124 samples that amplified the K1 subfamily showed a single allele of approximately 235 bp that was confirmed by sequencing. The same happened with the RO33 subfamily, with a single allele of approximately 335 bp. Only the MAD20 subfamily revealed the presence of three alleles with size polymorphism. No sequence polymorphisms were found within each allele. The only similar study conducted with 35 samples from Honduras and collected a decade ago described a total of 23 alleles (13 K1 alleles, 5 MAD20 alleles, and 5 RO33 alleles) [17]. This inconsistency reflects a bottleneck effect in the parasite populations suffered by the notable decrease in the number of cases in the region. This bottleneck led to decreased allelic richness, the near fixation of a few alleles and missing pre-existing alleles.

Despite the homogeneity in the population demonstrated in this study, 96/147 (65.3%) samples showed a polyclonal infection with at least two alleles of *pfmsp*-1, especially K1/RO33. This proportion is higher than previously reported in Honduras, where 23% of mixed samples are described [17]. Mixed infections were not detected for the *pfmsp*-2 gene. If we consider the recent bottleneck, the homogeneity of the mixed genotypes and the low transmission that continues to prevail in the region, it is very likely that these mixed genotypes are propagated by a single multiple inoculation event rather than by superinfection [40]. This hypothesis is consistent with the results obtained by Valdivia et al. [22] when analyzing 20 samples collected between 2008 and 2010 from Honduras, demonstrating exclusively monoclonal infections and low intrapopulation diversity using a genome sequencing approach.

Finally, the allele frequencies of the parasite populations of Honduras were compared with those of Nicaragua. Statistical analysis showed that there were no significant differences between the two populations. Our results are consistent with the results of a previous study conducted with 110 samples collected in the Moskitia (Honduras and Nicaragua) between 2009 and 2012 that evaluated seven neutral microsatellite markers [21]. Those authors determined that there was low genetic diversity in the population, similar allelic richness in both countries, and no significant differences in the number of alleles per locus between the two countries. Therefore, it was proposed that there was a relatively free gene flow across the region. Both results confirm that despite the political border between both countries, the parasites that circulate in the Moskitia region are not geographically separated, and that adverse genetic events such as the emergence of resistance to antimalarials would affect malaria control on both sides of the border.

## 4. Materials and Methods

### 4.1. Sample Collection

This study included 160 blood samples from febrile patients diagnosed with falciparum malaria and who sought medical assistance in national health establishments in two municipalities of Honduras and five municipalities of Nicaragua belonging to the ecological and cultural region called the Moskitia (Figure 7). All samples from Honduras were collected in the department of Gracias a Dios, and those from Nicaragua were collected in the North Atlantic Region (RAAN). This study was limited to studying cases of falciparum malaria and not the cases of vivax malaria because the Health Ministries of Honduras and Nicaragua routinely collect on filter paper only blood samples diagnosed with *P. falciparum*. The samples were collected between 2018 and 2021.

Blood samples were collected by fingerstick on Whatman FTA filter paper at the time of patient recruitment. The ethics committee (CEI-MEIZ) of the National Autonomous University of Honduras (UNAH) reviewed and approved the study under protocol number 03-2020. Consent to participate was waived for the following reasons: (a) No personal information was included; (b) the study was beneficial to public health and, (c) did not harm the participants. Blood samples were collected for parasite species identification and analysis of genes associated with drug resistance, in accordance with national regulations and for routine malaria surveillance purposes.

### 4.2. DNA Extraction and Molecular Identification of the Parasite

DNA was extracted from blood on filter paper cards using a Chelex-100 based method (Bio-Rad Laboratories, Inc., Hercules, CA, USA) [41]. Microscopic diagnosis of the parasite species was confirmed by amplification of the 18S rRNA gene [42].

### 4.3. Amplification of pfmsp-1 and pfmsp-2

The *pfmsp*-1 gene was amplified by nested PCR as described previously [17,43]. In the first PCR, a region of more than 1 kb of the second variable region of the gene was amplified and three independent nested amplifications were carried out to detect the allelic subfamilies K1, RO33 and MAD20 (Figure 8). Primary reactions were carried out in a 50 μL volume containing 25 μL of GoTaq^®^ 2X Master Mix (Promega corp., Madison, WI, USA), 2 μL of each primer (Table 3) at a concentration of 10 μM, 10 μL of DNA and 11 μL of nuclease-free water. Thermal cycler conditions were 1 cycle of 94 °C for 2 min; 30 cycles at 94 °C for 30 s, 54 °C for 1 min and 72 °C for 1 min; and a final extension step at 72 °C for 5 min. Two µL of the first PCR amplicons were used in three independent reactions in a 50 µL volume each containing 25 µL of Taq Master Mix, 2 µL of each primer (10 µM) and 19 µL of nuclease-free water. The cycling conditions in the thermal cycler for the three reactions were: 1 cycle of 94 °C for 5 min; 35 cycles at 94 °C for 30 s, annealing for 1 min and 72 °C for 1 min; and 1 final extension cycle at 72 °C for 10 min.

The *pfmsp*-2 gene was amplified by semi-nested PCR [17,44]. The first reaction was carried out in a 50 µL reaction volume containing 25 µL of Taq Master Mix, 2 µL of each primer (10 µM) (Table 3), 16 µL of nuclease-free water and 5 µL of DNA. The cycle conditions were: 1 denaturing step at 95 °C for 5 min; 30 cycles at 94 °C for 1 min, 60 °C for 2 min and 72 °C for 2 min; a final extension step at 72 °C and 10 min. The two allelic subfamilies 3D7 and FC27 were detected in independent secondary reactions in a final volume of 50 μL, with 25 μL of Taq Master Mix, 2 μL of each primer (10 μM), 20 μL of nuclease-free water and 1 μL DNA from the first reaction. The same cycling program was used for both reactions: 1 cycle at 94 °C for 2 min; 30 cycles at 94 °C for 30 s, 50 °C for 45 s, 70 °C for 2 min; and a final extension at 70 °C for 10 min. Amplification products were analysed by electrophoresis on a 2% agarose gel stained with ethidium bromide.

The overall allele frequency was calculated for each marker and the allele frequency in each of the two countries, Honduras, and Nicaragua. Pearson’s chi-square test was used to evaluate the percentages of positivity for each genotype per country to determine significant differences. A representative sample of each allele was selected according to the size of the product on the agarose gel for subsequent sequencing. Sequencing of amplicons was carried out directly from purified PCR products on both strands with their respective inner primers following standard Sanger sequencing protocols at Psomagen facilities (Psomagen, Inc., Rockville, MA, USA).

Sequences were trimmed at both 5′ and 3′ ends with the Geneious^®^ 9.1.7 software (Biomatters, Auckland, New Zealand) and queried against international databases contained in NCBI to confirm the identity of the sequences. Subsequently, the sequences were analyzed for sequence and/or size polymorphisms. The sequences obtained were deposited in the NCBI database. The nucleotide sequences were translated in silico in the correct open reading frame to obtain the amino acid sequences of K1, MAD20, RO33, 3D7 and FC27. The peptide sequences were analyzed with the NCBI Protein Blast tool.

## 5. Conclusions

This study describes a bottleneck effect suffered by *Plasmodium falciparum* populations in the Moskitia region of Central America. The drastic decrease in the number of malaria cases reported in the region in recent years could have been responsible for a notable decrease in the genetic diversity of the parasite, generating highly homogeneous populations as revealed by the analysis of the *pfmsp*-1 and *pfmsp*-2 genes. These results will be useful for future epidemiological investigations and the monitoring of malaria transmission in Central America.

## Figures and Tables

**Figure 1 pathogens-10-01432-f001:**
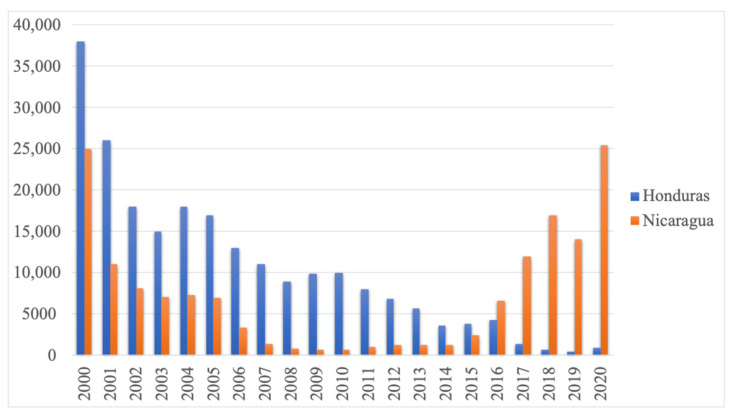
Number of malaria cases in Honduras and Nicaragua from 2000 to 2020. Source: Ministries of Health of Honduras and Nicaragua.

**Figure 2 pathogens-10-01432-f002:**
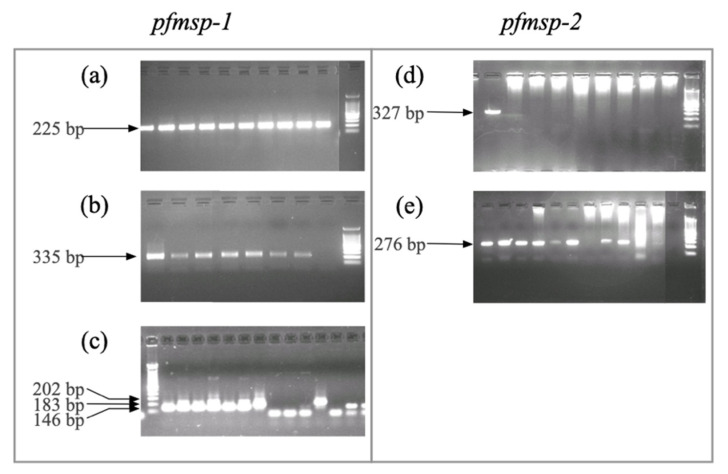
Agarose gels showing the amplification bands of the allelic subfamilies of *pfmsp*-1 and *pfmsp*-2. (**a**) K1; (**b**) RO33; (**c**) MAD20; (**d**) FC27; (**e**) 3D7. The molecular weight marker indicates a range of 100 bp to 1500 bp.

**Figure 3 pathogens-10-01432-f003:**
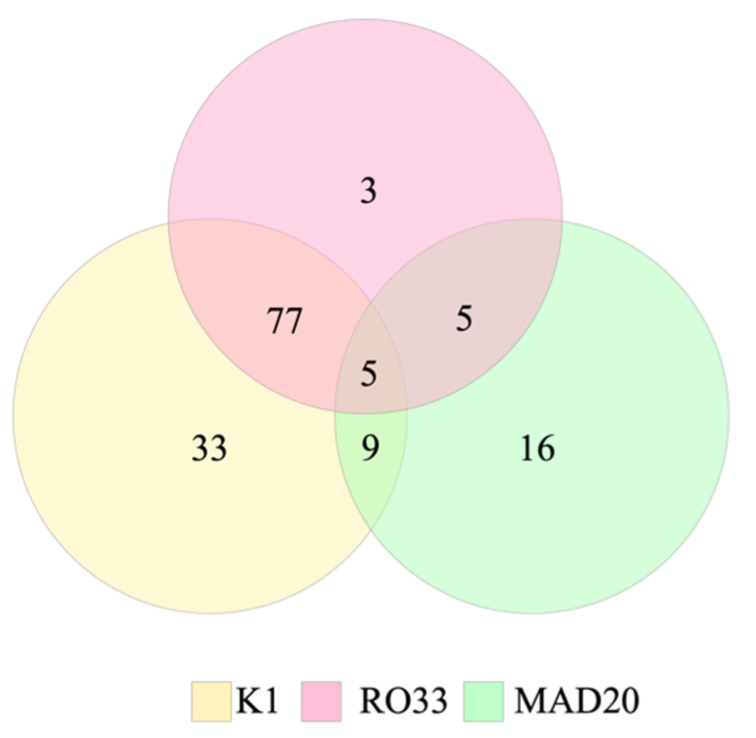
Venn diagram showing the number of samples that amplified for each allelic subfamily of *pfmsp*-1.

**Figure 4 pathogens-10-01432-f004:**
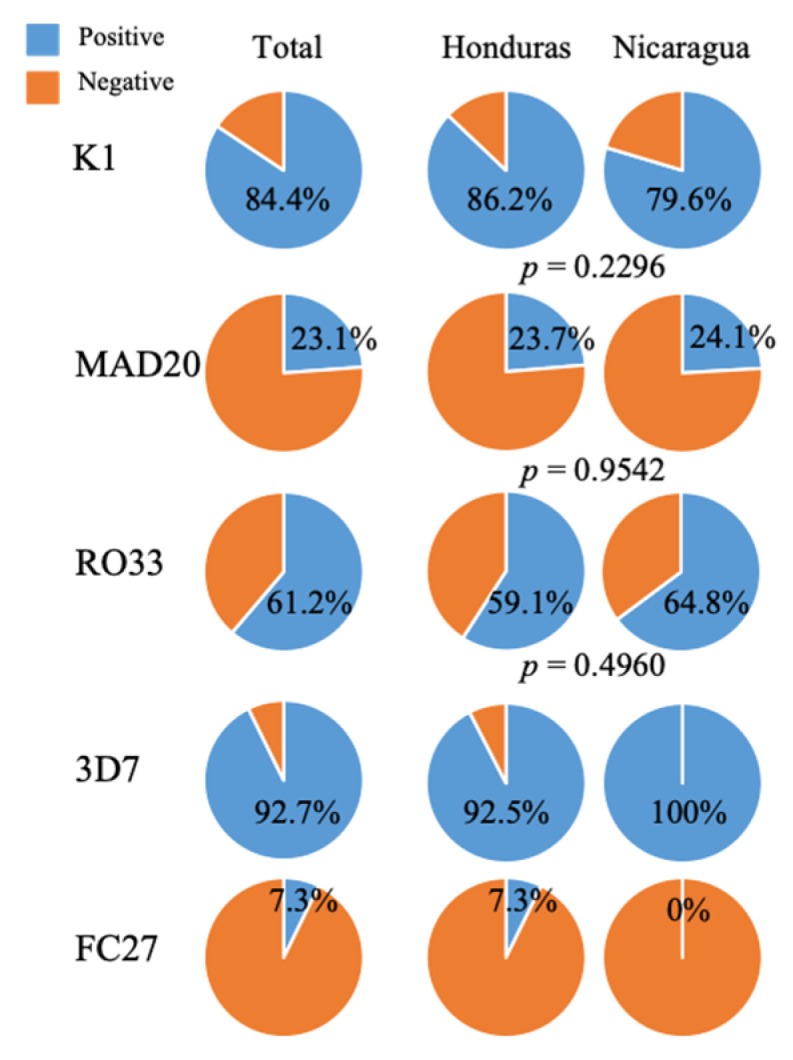
Proportion of *Plasmodium falciparum* isolates belonging to each of the allelic subfamilies of *pfmsp*-1 and *pfmsp*-2. The result of the chi-square test that compares the distribution of the three families of *pfmsp*-1 in Honduras and Nicaragua is shown.

**Figure 5 pathogens-10-01432-f005:**
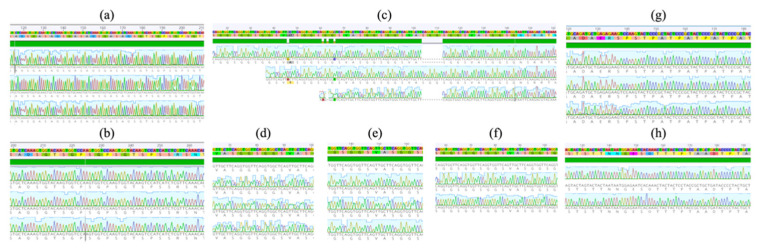
Chromatograms showing partial nucleotide sequence alignments of (**a**) K1; (**b**) RO33; (**c**) three different alleles of MAD20; (**d**) MAD20 type I; (**e**) MAD20 type II; (**f**) MAD20 type III; (**g**) 3D7; (**h**) FC27. The solid green bar above each alignment indicates that the aligned sequences do not show any difference between them.

**Figure 6 pathogens-10-01432-f006:**
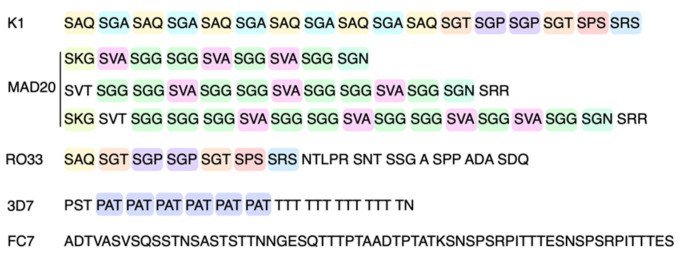
Amino acid sequences of the variable regions of each allelic subfamily of *pfmsp*-1 and *pfmsp*-2. Colors indicate different blocks of three amino acids.

**Figure 7 pathogens-10-01432-f007:**
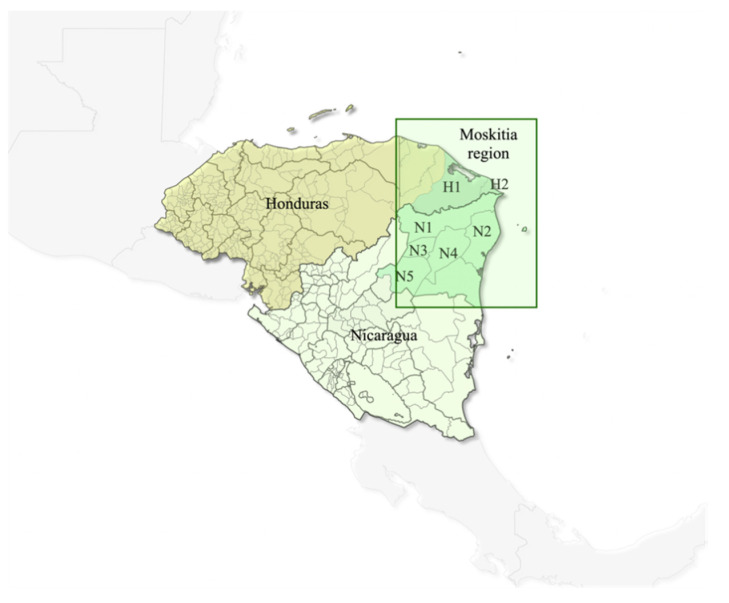
Map of Honduras and Nicaragua showing the municipalities in which blood samples were collected from patients with falciparum malaria. H1 = Puerto Lempira; H2 = Ramón Villeda Morales; N1 = Waspán; N2 = Puerto Cabezas; N3 = Bonanza; N4 = Rosita; N5 = Siuna, where H stands for Honduras and N for Nicaragua.

**Figure 8 pathogens-10-01432-f008:**
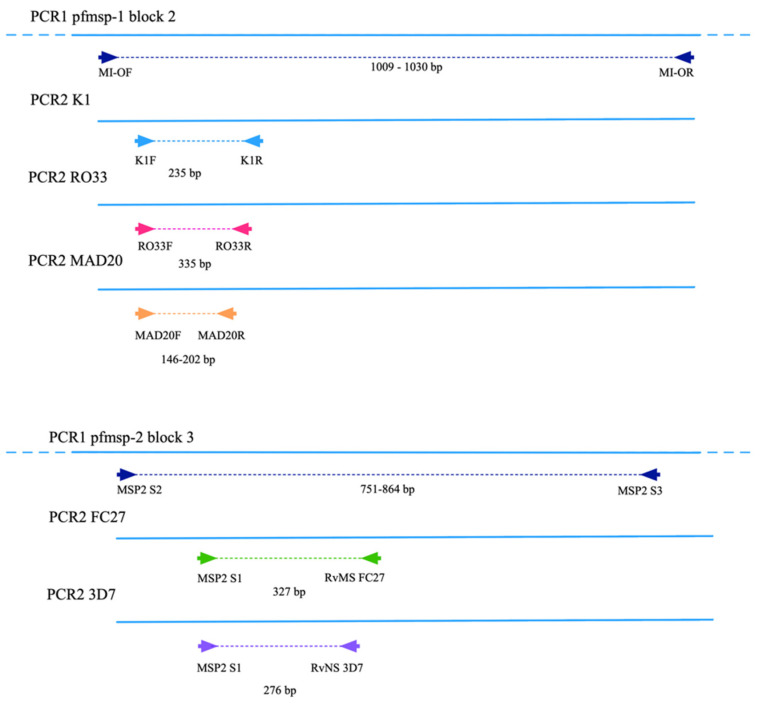
Scheme of *pfmsp*-1 and *pfmsp*-2 genes showing the size of the amplicons and hybridization sites of the primers.

**Table 1 pathogens-10-01432-t001:** Genotypes and allelic frequencies of *Plasmodium falciparum* according to the subfamilies of *pfmsp*-1 and *pfmsp*-2.

*pfmsp*-1 Subfamilies						*pfmsp*-2 Subfamilies			
K1 (bp)	n (%)	MAD20 (bp)	n (%)	RO33 (bp)	n (%)	3D7 (bp)	n (%)	FC27 (bp)	n (%)
235	124 (84.4)	146	6 (4.01)	335	90 (61.2)	276	51 (92.7)	327	4 (7.3)
Negative	23 (15.6)	183	16 (10.9)	Negative	57 (38.8)	Negative	4 (7.3)	Negative	51 (92.7)
		202	13 (8.8)						
		Negative	112 (76.2)						
Total	147	Total	147	Total	147	Total	55	Total	55

**Table 2 pathogens-10-01432-t002:** Frequency (%) of allelic subfamilies of *pfmsp*-1 and *pfmsp*-2 per country.

	*pfmsp*-1 Subfamilies							*pfmsp*-2 Subfamilies	
	K1	MAD20	RO33	K1/MAD20	MAD20/RO33	K1/RO33	K1/MAD20/RO33	3D7	FC27
Honduras	81	22	55	8	0	48	4	49	4
Nicaragua	43	13	35	1	5	29	1	2	0
Total	124 (84.4%)	35 (23.8%)	90 (61.2%)	9 (6.1%)	5 (3.4%)	77 (52.4%)	5 (3.4%)	51 (92.7%)	4 (7.3%)

**Table 3 pathogens-10-01432-t003:** Primers and amplification conditions for the *pfmsp*-1 and *pfmsp*-2 genes. A single band of the same size was found in all samples for each allelic family, except for MAD20.

	PCR/Allelic Family	Primer Name	Primer Sequence 5′-3′	Annealing Temperature	Approximate Amplicon Size (bp)
*pfmsp*-1	1st	MI-OF	CTAGAAGCTTTAGAAGATGCAGTATTG	54 °C	>1000
		MI-OR	CTTAAATAGATTCTAATTCAAGTGGATCA		
	2nd K1	K1F	AAATGAAGAAGAAATTACTACAAAAGGTGC	59 °C	235
		K1R	GCTTGCATCAGCTGGAGGGCTTGCACCAGA		
	2nd RO33	RO33F	TAAAGGATGGAGCAAATACTCAAGTTGTTG	58 °C	335
		RO33R	CAAGTAATTTTGAACTCTATGTTTTAAATCAGCGTA		
	2nd MAD20	MAD20F	AAATGAAGGAACAAGTGGAACAGCTGTTAC	59 °C	146–202
		MAD20R	ATCTGAAGGATTTGTACGTCTTGAATTACC		
*pfmsp*-2	1st	MSP2 S2	GAAGGTAATTAAAACATTGTC	60 °C	905
		MSP2 S3	GAGGGATGTTGCTGCTCCACAG		
	2nd 3D7	RvNS 3D7	CTGAAGAGGTACTGGTAGA	50 °C	276
		MSP-2 S1	GCTTATAATATGAGTATAAGGAGAA		
	2nd FC27	RvMS FC27	GCATTGCCAGAACTTGAA	50 °C	327
		MSP-2 S1	GCTTATAATATGAGTATAAGGAGAA		

## Data Availability

Data are contained within the article.

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
