# Peer review of "Evidence of a Recent Bottleneck in Plasmodium falciparum Populations on the Honduran–Nicaraguan Border"

_pathogens, 2021, doi:10.3390/pathogens10111432_

Round 1

Reviewer 1 Report

The manuscript examines samples from the Moskitia region in Central America for the presence of P. falciparum and the characteristics of these parasites. They use 2 genes with high polymorphism to assess parasite genetic diversity and the clonality of the infections. Comparing their results with previous studies, the authors assess the prevalence and transmission in the area.

Overall, the manuscript presents interesting data and contributes to the understanding of the status of malaria in Central America. It is very well written but there are some aspects that need to be made clearer.

The introduction states that Central America has low transmission and that studies using different methods show low diversity in the parasite population, compared to other regions of the continent and the world. But there are no references of regions in the American continent being classified as high transmission. This sentence could do with clarification because though it is widely understood that sub-Saharan Africa has high malaria transmission. This is much less clear in the Americas, where some of the regions with high incidence of malaria are still within the low to medium transmission levels and accordingly show low diversity of the parasite populations, as reported by Knudson et al, Sci Rep 2020.

There are several references to a recent bottleneck phenomenon in the region but it is not explained what this was. There is a hint at the dramatic decrease in infection rates in 2014, but more detail would be helpful.

The general characteristics of the Moskitia are described and it is presented as a region on the border of Honduras with Nicaragua. The wording portrays this region as being in Honduras on the border with Nicaragua and though the map of the region and the collection points is in the methods section at the end of the manuscript, there is no reference to this when the results are presented. Together with no information being given about human migration in the area, table 2 presenting the cases per country comes as a bit of a surprise because there is no reference to samples being collected in Nicaragua as well as Honduras or the presence of Nicaraguans in the region. It would be clearer for the reader to either include the map as a first figure in the results section or make a clear reference to it. Also, though the geographical region and the localities of sample collection are shown on the map, the number of samples collected in each locality in each year of collection is not specified. The map itself lacks detail, such as the border between the 2 countries, the names of the Departments of Honduras, etc that would allow to put into context the data presented.

               The results section states that 160 samples were taken between 2018 and 2021. Of these, 147 amplified for at least 1 of the three allelic subfamilies of pfmsp-1 block 2: K1 84.4%, MAD20 23.8% and RO33 61.2%. 55 samples amplified for at least one family of pf msp2 block 3: 3D7 92.7% and FC27 7.3%. So what happened with the rest of the samples that didn’t amplify? Were they P. vivax? Or an unknown P. falciparum allele? Or was there insufficient material in the samples?

               Figure 2 needs more detail, at least the lowest MW standards should be indicated to give guidance to the molecular weight of the bands. Also, the description of the results states that no intrafamily multiclonal infections were found, yet at least two lanes with a doublet can be seen in the MAD20 gel presented, that presumably come from the same individual samples. The MW standards in some of these gels clearly come from another gel, when this is the case there should be a separation between the different pieces of gel.

               Figure 6 (not to be confused the map which is also figure 6!) could be included as part B of figure 5 and should have a key to understand the colour coding of the aminoacid sequences. It is not clear what the variation is because the text states there is no diversity in the sequences other than the three alleles of MAD20, presumably this refers to the difference between them rather than each presenting variation. Then it says that the aminoacid sequences were analysed by blast and the percentages of identity were up to 100%... what does this mean? That can be 1% or even 0%, it needs to be specified what level of identity to previously described sequences was found, which would be quite important given that the study makes conclusions about the state of transmission over time comparing to previous data. The chromatograms are quite difficult to follow, perhaps it would be much better to present the alignment of the nucleotides of all the sequences rather than some examples with the profiles.

The discussion is quite superficial and a more in depth analysis of the data and the context of previous studies would be beneficial to increase the insight of the manuscript. For example the geographical regions where samples were collected vary between the studies and indeed, the only one study reporting high diversity in the P. falciparum population is also the only one that analysed samples originating from departments along the border with Nicaragua as well as departments on the northern Caribbean coast including the islands. The geographical characteristics where samples originate are certainly important but the features of the regions, the style of the communities and the movement of people are crucial in understanding the evolution of parasite diversity and changes in the dynamics of transmission.

The conclusion states that the manuscript describes a bottleneck effect in the parasite population leading to lower genetic diversity due to the decrease in prevalence and transmission. However the genetic diversity over time is not very clearly presented, little attention is devoted to multiclonal infections and the study is based on sequences of regions of 2 genes, which is a good proxi for variation but it’s not the full story. This part of the discussion and conclusions should be improved and clarified.

Minor points

  • It would be helpful for setting the scenario to justify why the study concentrated on P. falciparum.
  • What does it mean that amplicons were successfully sequenced and assembled? None of them was much bigger than 1 kb
  • Following the discussion that the dynamics of parasite populations have changed over time, maybe it would be interesting to see whether there are some differences in the times of sampling in this study ie 2018 to 2021, if numbers of samples allow.
  • As mentioned above, earlier studies referred to in lines 173-78 should be placed into the geographical context relative to the present work.
  • Sentence 178, was RO33 not investigated or not found in this study?
  • sentence 182 should state which Latin American countries, in order to have a geographical perspective of the distribution of genotypes and clonal characteristics of parasite populations.
  • Reference 13 is a study from 1999 that collected samples from 2 communities in the Department of Colon, on the northern Caribbean coast, not 1 city. Line 178
  • Though reasons for the consent waiver are mentioned, these are not very clear. Any study is expected to cause no harm to the participants and aim to benefit the public and the fact that no personal information was included does not preclude the necessity of consent to a sample being taken. Also, though personal information might not be included in the study at least the domicile and origin of the volunteers has to be recorded?.
  • The conclusions would be better after the discussion that at the end of the manuscript.

Author Response

The manuscript examines samples from the Moskitia region in Central America for the presence of P. falciparum and the characteristics of these parasites. They use 2 genes with high polymorphism to assess parasite genetic diversity and the clonality of the infections. Comparing their results with previous studies, the authors assess the prevalence and transmission in the area.

Overall, the manuscript presents interesting data and contributes to the understanding of the status of malaria in Central America. It is very well written but there are some aspects that need to be made clearer.

  1. The authors would like to deeply appreciate the reviewer's full and detailed comments and observations. We have tried to satisfy all doubts and correct errors in the hope of improving the scientific quality of the manuscript.
  2. The introduction states that Central America has low transmission and that studies using different methods show low diversity in the parasite population, compared to other regions of the continent and the world. But there are no references of regions in the American continent being classified as high transmission. This sentence could do with clarification because though it is widely understood that sub-Saharan Africa has high malaria transmission. This is much less clear in the Americas, where some of the regions with high incidence of malaria are still within the low to medium transmission levels and accordingly show low diversity of the parasite populations, as reported by Knudson et al, Sci Rep 2020.

R/ The paragraph has been modified as follows for clarity and precision; (however, it is not possible to elaborate on this idea in the introduction): “Despite differences in the methods, most of these studies conclude that the parasite populations in Central America present a low diversity in relation to parasites of Sub-Saharan Africa [21]. This is much less clear in the Americas, where some of the regions with high incidence of malaria are still within the low to medium transmission levels and accordingly show low diversity of the parasite populations [22].”

  1. There are several references to a recent bottleneck phenomenon in the region but it is not explained what this was. There is a hint at the dramatic decrease in infection rates in 2014, but more detail would be helpful.
  • R/ The first reference to a “bottleneck effect” in the manuscript occurs in the abstract. The wording has been enriched as follows for a better reader understanding: “The continuous decrease in cases between 2000 and 2014, followed by a rapid expansion from 2015 to the present, has generated a potential bottleneck effect in the populations of Plasmodium
  • The second time the bottleneck effect is mentioned is at lines 56-58; However, in that same paragraph, the statistical data describe the way in which the number of cases decreased from 2000 to 2014 and has subsequently increased. These data are visually supported by figure 1, so that even if the reader is not an expert in population genetics, they can easily understand the phenomenon. So there is more than a hint of the phenomenon.
  • The third mention occurs in the final paragraph of the introduction in which the objective of the study is described: “This study aimed to analyze the changes in the genetic diversity of P. falciparum population circulating in the Moskitia after a recent bottleneck phenomenon”.
  • In the discussion it is said (lines 171-2) that a possible bottleneck effect, added to the recent change in the transmission of the two species of the parasite, justifies the study of the genetic diversity of the parasite.
  • Lines 207-8 comment that the notable decrease in allelic richness in pfmsp-1 in recent years would be evidence of a bottleneck effect.

  1. The general characteristics of the Moskitia are described and it is presented as a region on the border of Honduras with Nicaragua. The wording portrays this region as being in Honduras on the border with Nicaragua and though the map of the region and the collection points is in the methods section at the end of the manuscript, there is no reference to this when the results are presented. Together with no information being given about human migration in the area, table 2 presenting the cases per country comes as a bit of a surprise because there is no reference to samples being collected in Nicaragua as well as Honduras or the presence of Nicaraguans in the region. It would be clearer for the reader to either include the map as a first figure in the results section or make a clear reference to it. Also, though the geographical region and the localities of sample collection are shown on the map, the number of samples collected in each locality in each year of collection is not specified. The map itself lacks detail, such as the border between the 2 countries, the names of the Departments of Honduras, etc that would allow to put into context the data presented.
  • R/ To help the reader better understand the location of the Miskito region, the paragraph between lines 47 and 51 has been modified as follows: “Currently the Central American region most affected by malaria is the Moskitia, which comprises the easternmost region of Honduras and the northwestern region of Nicaragua. The Moskitia is a region shared by both countries with little human development, scarce communication routes and predominantly indigenous populations with their own cultural characteristics and little access to health services.”
  • We understand the reviewer's surprise regarding the results in Table 2; however, the format of the Pathogens journal requires that materials and methods be placed at the end of the document.
  • Figure 6 has been modified to clearly indicate the separation between the two countries. In addition, the approximate region that includes Moskitia has been framed.
  • To specify the origin of the samples from both countries, the following sentence has been included in the first paragraph of materials and methods: “All samples from Honduras were collected in the department of Gracias a Dios, and those from Nicaragua were collected in the North Atlantic Region (RAAN).”
  • It is true that the number of samples collected in each locality in each year of collection is not specified. However, the authors did not believe it necessary to include this information given the high genetic homogeneity between municipalities. However, all the information can be found in the supplementary material that has been included.

               The results section states that 160 samples were taken between 2018 and 2021. Of these, 147 amplified for at least 1 of the three allelic subfamilies of pfmsp-1 block 2: K1 84.4%, MAD20 23.8% and RO33 61.2%. 55 samples amplified for at least one family of pf msp2 block 3: 3D7 92.7% and FC27 7.3%. So what happened with the rest of the samples that didn’t amplify? Were they P. vivax? Or an unknown P. falciparum allele? Or was there insufficient material in the samples?

R/ The cause of the lack of amplification of the pfmsp-1 gene in 13 samples is unknown. This clarification has been included in line 98.

               Figure 2 needs more detail, at least the lowest MW standards should be indicated to give guidance to the molecular weight of the bands. Also, the description of the results states that no intrafamily multiclonal infections were found, yet at least two lanes with a doublet can be seen in the MAD20 gel presented, that presumably come from the same individual samples. The MW standards in some of these gels clearly come from another gel, when this is the case there should be a separation between the different pieces of gel.

  • R/ The molecular weight marker indicates a range of 100 bp to 1500 bp. This has been included in the description of figure 2.
  • What the reviewer interprets as a band of less than 100 bp in the MAD20 subfamily is a primer dimer and not a band from an actual allele.
  • In Figure 2 (a) and 2 (e) a separation is observed between the molecular weight marker line and the rest of the gel, however it is the same gel and not a different gel. The lanes have been joined for aesthetic reasons and to avoid displaying non-informative spaces that could confuse the reader.

Figure 6 (not to be confused the map which is also figure 6!) could be included as part B of figure 5 and should have a key to understand the colour coding of the aminoacid sequences. It is not clear what the variation is because the text states there is no diversity in the sequences other than the three alleles of MAD20, presumably this refers to the difference between them rather than each presenting variation. Then it says that the aminoacid sequences were analysed by blast and the percentages of identity were up to 100%... what does this mean? That can be 1% or even 0%, it needs to be specified what level of identity to previously described sequences was found, which would be quite important given that the study makes conclusions about the state of transmission over time comparing to previous data. The chromatograms are quite difficult to follow, perhaps it would be much better to present the alignment of the nucleotides of all the sequences rather than some examples with the profiles.

  • R/ Joining Figures 5 and 6 would be an option, but we prefer to keep them separate for clarity in communicating the data.
  • The following has been indicated in the legend of figure 6: “Colors indicate different blocks of three amino acids.”
  • Figure 6 shows the unique amino acid sequence found for the K1, RO33, 3D7 and FC7 subfamilies, as well as the three sequences of the three alleles of the MAD20 family. We don't understand the reviewer's confusion on this.
  • The sentence on lines 147-150 has been modified as follows: “These sequences were analyzed by the Protein Blast tool and the results indicated identity percentages of 93.2% for RO33 and 100% for K1, MAD20 and 3D7, with respect to sequences previously deposited in the database of isolates of various geographic regions”.
  • Figure 5 shows the alignment of nucleotide sequences with an example of three sequences each. The following sentence has been included in the legend of figure 5: “The solid green bar above each alignment indicates that the aligned sequences do not show any difference between them”.

The discussion is quite superficial and a more in depth analysis of the data and the context of previous studies would be beneficial to increase the insight of the manuscript. For example the geographical regions where samples were collected vary between the studies and indeed, the only one study reporting high diversity in the P. falciparum population is also the only one that analysed samples originating from departments along the border with Nicaragua as well as departments on the northern Caribbean coast including the islands. The geographical characteristics where samples originate are certainly important but the features of the regions, the style of the communities and the movement of people are crucial in understanding the evolution of parasite diversity and changes in the dynamics of transmission.

R/ We are very sorry that the reviewer considers that our discussion is superficial, however we consider that our results have been compared with all the existing literature in the country and that it has been generated by our research group during the last 12 years. The reviewer suggests that a segmented comparison should be made by geographical subregions, considering the characteristics of each community, however this analysis, although interesting, far exceeds the objectives of this study. Added to this, it would not provide more data than that already provided because the entire Miskito region and the Caribbean of Honduras and Nicaragua share climatic and some socio-cultural conditions. Finally, the homogeneity in the genetic diversity of the parasites analyzed in this study makes it unnecessary to carry out further comparisons between localities.

The conclusion states that the manuscript describes a bottleneck effect in the parasite population leading to lower genetic diversity due to the decrease in prevalence and transmission. However the genetic diversity over time is not very clearly presented, little attention is devoted to multiclonal infections and the study is based on sequences of regions of 2 genes, which is a good proxi for variation but it’s not the full story. This part of the discussion and conclusions should be improved and clarified.

R/ Of course, the results obtained do not count "the full story". It is not the objective of this study to clarify the totality of the phenomenon behind the decrease in the genetic diversity of the parasite, but to reveal information based on evidence using two markers universally used for this purpose. Despite the limitations of the techniques available in Honduras, the results clearly reflect a possible bottleneck phenomenon in Pf populations in relation to the information obtained in previous years.

Minor points

It would be helpful for setting the scenario to justify why the study concentrated on P. falciparum.

R/ The following statement has been added in M&M: “This study was limited to studying the cases of falciparum malaria and not the cases of vivax malaria because the Health Ministries of Honduras and Nicaragua routinely collect on filter paper only blood samples diagnosed with P. falciparum.

What does it mean that amplicons were successfully sequenced and assembled? None of them was much bigger than 1 kb

R/ With “assembled” we refer to the union of the sequences obtained with both primers forward and reverse.

Following the discussion that the dynamics of parasite populations have changed over time, maybe it would be interesting to see whether there are some differences in the times of sampling in this study ie 2018 to 2021, if numbers of samples allow.

R/ There are no obvious differences between the samples for each year. This is the reason why the manuscript presents the data without distinction of year. For more information, the supplementary material that includes the database with all the results obtained has been made available to the Editor.

As mentioned above, earlier studies referred to in lines 173-78 should be placed into the geographical context relative to the present work.

R/ The paragraph has been modified as follows: “The genetic diversity of P. falciparum populations is directly proportional to the in-tensity of malaria transmission in a geographic region [6]. High genetic diversity of the parasite is characteristic of areas of highest malaria transmission while diversity is expected to be minimal in low transmission settings. For example, a recent study conducted in Northeast India using merozoite surface protein 1 and 2 (pfmsp-1, pfmsp-2) and glutamate rich protein (pfglurp) revealed extensive parasite diversity in a state that con-tributes almost half of malaria cases to the region [26]. An earlier study demonstrated dramatic variation in geographic diversity and differentiation in different regions using microsatellite analysis. The mean heterozygosities in South American countries were less than half of those observed in Africa, with intermediate heterozygosities in the Asia / Pacific region [33].”

Sentence 178, was RO33 not investigated or not found in this study?

R/ Line 278? The reference literally describes that: “The pfmsp-1 sequences from different geographic areas of Myanmar, India, Thailand, Vietnam, Philippines, Papua New Guinea, Solomon Islands, Vanuatu, Ghana,

Kenya, Tanzania, Uganda, Peru, and Brazil were comparatively analysed, to understand the genetic structure of pfmsp-1 in the global P. falciparum population. Although the K1 and MAD20 types were the dominant ones in

global pfmsp-1, the overall prevalence and distribution of pfmsp-1 allelic types differed by country or continent (Fig. 4).”

Sentence 182 should state which Latin American countries, in order to have a geographical perspective of the distribution of genotypes and clonal characteristics of parasite populations.

R/ Line 282? Modified as follows: “In two Latin American countries (Peru and Brazil), K1 shows a slight predominance over MAD20, while the frequency of RO33 is only marginal”.

Reference 13 is a study from 1999 that collected samples from 2 communities in the Department of Colon, on the northern Caribbean coast, not 1 city. Line 178

R/ That is correct. The term “city” has been changed by “municipality”.

Though reasons for the consent waiver are mentioned, these are not very clear. Any study is expected to cause no harm to the participants and aim to benefit the public and the fact that no personal information was included does not preclude the necessity of consent to a sample being taken. Also, though personal information might not be included in the study at least the domicile and origin of the volunteers has to be recorded?

R/ No personal data of the participants was available to the researchers, since the samples were collected by officials of the Ministry of Health for routine diagnosis and surveillance of malaria, and the samples arrived at our laboratory coded and anonymized. Only the municipality of origin of the patient was consigned.

  • The conclusions would be better after the discussion that at the end of the manuscript.

R/ We follow the format indicated by Pathogens journal.

Reviewer 2 Report

The authors analyzed the genetic diversity of P. falciparum populations, based on msp-1 and msp-2, in 160 samples collected in 2018–2021 in Nicaraguan-Honduran border area. In this region, after a decade of declining incidence, a rebound of incidence, mostly due to P. falciparum rather than P. vivax, has been reported in recent years. The present work seems to be a follow-up study of the paper Larranaga et al 2013 (Ref 19).

The introduction is informative and adequate. The methods are clearly described in detail. PCR protocols were based on earlier published studies. Some of the samples (n = 51) were sequenced, and the sequence data were deposited in GenBank. The results showed a very limited degree of polymorphism (only in MAD20 allelic family) and support a bottleneck effect. Discussion is adequate and interesting.

MAJOR COMMENTS:

Fig. 1: The graph is better presented as a histogram because the numbers of malaria cases are discrete variables each year (i.e., they are not continuous variables). For example, the total number of cases in 2000-2020 is not obtained by calculating the area under the curve, but by adding annual number of cases in each country. The source of data (Ministries of Health?) should be cited.

Fig. 4: The same data are in the main text (line 121) and Table 2. This figure is redundant and should be deleted. P. falciparum, pfmsp-1, and pfmsp-2 in italics.

MINOR COMMENTS:

Introduction:

Lines 61-64, Ref 3-5: Although the statement (“the genetic diversity of Plasmodium spp….”) is correct, it would be preferable to cite works performed in South America. All three references cited here (Ref 3-5) are studies from Africa. The authors claim that “studies on the genetic diversity of Plasmodium spp. in Central America are scarce” (lines 68-69). In the literature, there should be studies on genetic diversity performed in South America.

Lines 64-66, Ref 6-10, “Genetically diverse populations of Plasmodium are common in regions with high transmission patterns, such as sub-Saharan Africa [6-10]…”:  Ref 7 is a work done in Malaysia. This reference does not support the statement concerning Africa. Ref 7 should be deleted here.

Line 70, abbreviations of genes (pfglurp, pvama-1, pvmsp-1…): The journal “Pathogens” is not a highly specialized journal for malaria researchers. It would be helpful for uninitiated readers to provide the meaning of the abbreviations in the main text. The authors can put some order in these cited genes, by enumerating P. falciparum genes first, followed by P. vivax genes. “pfmps-2” should be corrected to “pfmsp-2”.

Line 72, “genomic sequencing [20]”: Please provide more detail here. Which genes were studied? Is it by whole genome sequencing?

Line 84: A period (.) after “located on chromosome 2” then start a new sentence “It is composed of…”

Line 86: aimed

Results:

Lines 90-97: In this paragraph, the authors should state the number (and percentage) of successful PCR. If I understand correctly, the PCR success rates were 92% (147/160) and 34% (55/160) for msp1 and msp2, respectively. Please clarify.

Line 91: P. falciparum in italics (not “Plasmodium falciparum”)

Lines 92, 107, 109, 112, 132, Fig 2 caption, Fig 3 caption, Fig 6 caption: pfmsp-1 in italics

Lines 94, 108,132, Fig 2 caption, Fig 6 caption: pfmsp-2 in italics

Tables 1 and 2 can be improved. For further clarity, it should be shown that K1, MAD20, and RO33 are msp1 allelic families and 3D7 and FC27 are msp2 allelic families.

Line 105: 16 (10.9%)

Line 120: the relative distribution…does not show

Line 121: The authors should state that these p values are not statistically significant.

Fig. 5: The text is clear. These chromatograms can be presented as supplementary data.

Discussion:

Line 151: P. falciparum, in italics (not Plasmodium falciparum)

Lines 184-185: reported an almost absolute predominance of MAD20 and did not report the presence of RO33

Lines 185-187: Ref 25 is a work done in Myanmar (southeast Asia); Ref 34 in Yemen (Middle East). Please add a study performed in Africa to correctly support the authors’ statement.

Line 195: reported (instead of “reports”)

Line 207: low (instead of “high”) homogeneity?

Line 229: delete “it”

Methods:

Line 246: Blood samples were collected…

Line 252: 18S rRNA gene

Line 258: Please provide the exact composition (and manufacturer) of “Taq Master Mix.”

Line 291: Delete “looking”

Table 3: Due to size polymorphisms, the amplicon size is expected to vary with each sample. This is especially true for samples from other malaria endemic areas where Pf diversity has been observed. I think that it would be very helpful to state in the Table legend that only a single band of the same size was found in all samples for each allelic family, except for MAD20, in the authors’ sample.

Fig. 6 (map): Please check the figure number (it should be Figure 7). Please add in the figure legend that H stands for Honduras and N for Nicaragua.

Ref 12: The format is different from that of other references. The first letters of the article title are in capital letters. They should be in small letters, except for proper names and the first letter of the first word of the title.

Ref 22: Same comment as for Ref 12 – “Merozoite” and “Protein”

Ref 28: “Merozoite”

Author Response

REVIEWER 2

The authors analyzed the genetic diversity of P. falciparum populations, based on msp-1 and msp-2, in 160 samples collected in 2018–2021 in Nicaraguan-Honduran border area. In this region, after a decade of declining incidence, a rebound of incidence, mostly due to P. falciparum rather than P. vivax, has been reported in recent years. The present work seems to be a follow-up study of the paper Larranaga et al 2013 (Ref 19).

The introduction is informative and adequate. The methods are clearly described in detail. PCR protocols were based on earlier published studies. Some of the samples (n = 51) were sequenced, and the sequence data were deposited in GenBank. The results showed a very limited degree of polymorphism (only in MAD20 allelic family) and support a bottleneck effect. Discussion is adequate and interesting.

  1. The authors would like to deeply appreciate the reviewer's full and detailed comments and observations. We have tried to satisfy all doubts and correct errors in the hope of improving the scientific quality of the manuscript.

MAJOR COMMENTS:

Fig. 1: The graph is better presented as a histogram because the numbers of malaria cases are discrete variables each year (i.e., they are not continuous variables). For example, the total number of cases in 2000-2020 is not obtained by calculating the area under the curve, but by adding annual number of cases in each country. The source of data (Ministries of Health?) should be cited.

R/ We agree with the reviewer. The graph has been replaced by a histogram. The source of the data has been included in the legend of the figure.

Fig. 4: The same data are in the main text (line 121) and Table 2. This figure is redundant and should be deleted. P. falciparum, pfmsp-1, and pfmsp-2 in italics.

R/ The reviewer is correct. Most of the information contained in table 2 is expressed graphically in figure 4, adding the result of the chi-square analysis. Although figure 4 is not essential for the manuscript, we hope that the editor will allow us to maintain it so that the reader can more quickly and more comfortably perceive the similarities between countries.

MINOR COMMENTS:

Introduction:

 Lines 61-64, Ref 3-5: Although the statement (“the genetic diversity of Plasmodium spp….”) is correct, it would be preferable to cite works performed in South America. All three references cited here (Ref 3-5) are studies from Africa. The authors claim that “studies on the genetic diversity of Plasmodium spp. in Central America are scarce” (lines 68-69). In the literature, there should be studies on genetic diversity performed in South America.

  • R/ Two references have been inserted (Colombia, Brazil) that describe mutations associated with genetic resistance to antimalarials.
  • Indeed, there are several studies that report the genetic diversity of the parasite in South America, however, the objective of the manuscript is to compare the diversity found in the present study with previous studies carried out in Central America. Either way, the introduction is not the right section for an in-depth comparison that exhausts the topic.

Lines 64-66, Ref 6-10, “Genetically diverse populations of Plasmodium are common in regions with high transmission patterns, such as sub-Saharan Africa [6-10]…”:  Ref 7 is a work done in Malaysia. This reference does not support the statement concerning Africa. Ref 7 should be deleted here.

R/ It's right. We thank the reviewer for his observation. The sentence has been expanded as follows: “high transmission patterns, such as sub-Saharan Africa and some Southeast Asian countries”…

Line 70, abbreviations of genes (pfglurp, pvama-1, pvmsp-1…): The journal “Pathogens” is not a highly specialized journal for malaria researchers. It would be helpful for uninitiated readers to provide the meaning of the abbreviations in the main text. The authors can put some order in these cited genes, by enumerating P. falciparum genes first, followed by P. vivax genes. “pfmps-2” should be corrected to “pfmsp-2”.

 R/ The paragraph has been modified as follows: “Central America is considered a region of moderate to low malaria transmission, and studies on the genetic diversity of Plasmodium spp. in Central America are scarce. Most studies have analyzed polymorphic coding genes (P. falciparum glutamate rich protein pfglurp, merozoite surface proteins pfmsp-1 and pfmsp-2, P. vivax apical membrane antigen pvama-1, merozoite surface proteins pvmsp-1, pvmsp-142, circumsporozoite protein pvcsp, and the surface protein pvs48 / 45) to assess the diversity of P. falciparum and / or P. vivax in some countries of the isthmus”…

Line 72, “genomic sequencing [20]”: Please provide more detail here. Which genes were studied? Is it by whole genome sequencing?

 R/ The sentence has been modified as follows: “or next generation sequencing using a selectively amplified whole genome approach (sWGA)”

Line 84: A period (.) after “located on chromosome 2” then start a new sentence “It is composed of…”

 R/ Modified.

Line 86: aimed

R/Corrected.

Results:

Lines 90-97: In this paragraph, the authors should state the number (and percentage) of successful PCR. If I understand correctly, the PCR success rates were 92% (147/160) and 34% (55/160) for msp1 and msp2, respectively. Please clarify.

 R/ Done. Thanks.

Line 91: P. falciparum in italics (not “Plasmodium falciparum”)

 R/Done.

Lines 92, 107, 109, 112, 132, Fig 2 caption, Fig 3 caption, Fig 6 caption: pfmsp-1 in italics

 R/Corrected.

Lines 94, 108,132, Fig 2 caption, Fig 6 caption: pfmsp-2 in italics

 R/Corrected.

Tables 1 and 2 can be improved. For further clarity, it should be shown that K1, MAD20, and RO33 are msp1 allelic families and 3D7 and FC27 are msp2 allelic families.

 R/ Both tables have been modified to clarify the subfamilies of the msp1 and msp2 genes.

Line 105: 16 (10.9%)

 R/Done.

Line 120: the relative distribution…does not show

 R/ Unfortunately we do not understand this reviewer observation. In any case, a supplementary table with the complete study database has been included.

Line 121: The authors should state that these p values are not statistically significant.

 R/ The sentence has been modified: “As shown in Figure 4 and Table 2, the relative distribution of alleles in the samples collected in Honduras and Nicaragua are not statistically significant”

Fig. 5: The text is clear. These chromatograms can be presented as supplementary data.

 R/ Thanks.

Discussion:

 Line 151: P. falciparum, in italics (not Plasmodium falciparum)

 R/ Corrected.

Lines 184-185: reported an almost absolute predominance of MAD20 and did not report the presence of RO33

 R/ Changed, thanks.

Lines 185-187: Ref 25 is a work done in Myanmar (southeast Asia); Ref 34 in Yemen (Middle East). Please add a study performed in Africa to correctly support the authors’ statement.

 R/ Although it is true that the study by Le et al 2019 used samples from Myanmar, they also analyzed sequences from 4 continents (South America, Africa, Southeast Asia and Oceania) see Fig 4.

Line 195: reported (instead of “reports”)

R/ Corrected.

Line 207: low (instead of “high”) homogeneity?

 R/ The samples in our study proved to be highly homogeneous with each other. However the phrase has been changed: “Despite the homogeneity in the population demonstrated in this study”

Line 229: delete “it”

 R/ Deleted.

Methods:

Line 246: Blood samples were collected…

 R/ Deleted.

Line 252: 18S rRNA gene

 R/ Modified.

Line 258: Please provide the exact composition (and manufacturer) of “Taq Master Mix.”

 R/ Added.

Line 291: Delete “looking”

 R/ Deleted.

Table 3: Due to size polymorphisms, the amplicon size is expected to vary with each sample. This is especially true for samples from other malaria endemic areas where Pf diversity has been observed. I think that it would be very helpful to state in the Table legend that only a single band of the same size was found in all samples for each allelic family, except for MAD20, in the authors’ sample.

 R/ Added. Thanks.

Fig. 6 (map): Please check the figure number (it should be Figure 7). Please add in the figure legend that H stands for Honduras and N for Nicaragua.

 R/ Thank you very much. Figure numbers 7 and 8 have been corrected. The legend of the map has also been modified.

Ref 12: The format is different from that of other references. The first letters of the article title are in capital letters. They should be in small letters, except for proper names and the first letter of the first word of the title.

 R/ Corrected.

Ref 22: Same comment as for Ref 12 – “Merozoite” and “Protein”

 R/ Corrected.

Ref 28: “Merozoite”

R/ Corrected.

Round 2

Reviewer 1 Report

The authors addressed the comments made and improved the manuscript accordingly.

I feel the need to clarify comment number 7 concerning the discussion. The comment was not intended to suggest a comparison of subregions, but rather to point out that the only report of high genetic diversity in the parasite population included a broader geographic region of the country, compared to other studies including some of the same period. It is therefore not entirely ‘surprising to find low genetic diversity in the parasite population’ in the present study. It seems rather selective to use this work alone to support the hypothesis of a bottleneck caused by the decrease of malaria in the region, while the other studies that found low parasite diversity are not included in the discussion of this point. Whether diversity was assessed using particular markers or other methods such as microsatellites or SNP barcoding, the results reflect the structure of the parasite population in the region.

Author Response

We are very grateful to the reviewer for the enriching comments made to the discussion. To broaden the discussion, we have modified the paragraph as follows: “Both genes encode antigens with highly polymorphic regions that have been used extensively to study the epidemiology of malaria [27,28,36]. The most frequent allelic subfamily of pfmsp-1 was K1 (84.4%) followed by RO33 (61.2%) and MAD20 (23.8%). An earlier study using 30 samples collected from six municipalities in Honduras (two in the Moskitia region) between 2010 and 2011 found that K1 was the most prevalent subfamily (57%), followed by MAD20 (25%) and RO33 (17%) [17]. The authors reported that most alleles were evenly represented, and no differences were found between the Miskito region and the rest of the country. The study by López et al included a broader geographic region of the country compared to other studies from the same period and despite having analyzed only 30 isolates of P. falciparum, it is the only one that reports high genetic diversity in the parasite population in Honduras [17]. Another previous study analyzed 56 samples collected between 1995 and 1996 in one municipality in Honduras (Tocoa, Colón) and found that the most frequent allelic subfamily was MAD20 (73.2%) followed by K1 (46.4%). RO33 was not described on that occasion [15]. These discrepancies can be attributed to the population dynamics of the parasite over time, or to the difference in the geographic areas where the samples were collected.”
